# Peer review of "Inhibition of Surface Corrosion Behavior of Zinc-Iron Alloy by Silicate Passivation"

_coatings, doi:10.3390/coatings13061057_

Round 1

Reviewer 1 Report

In this article, the authors used both a silicate passivation system and a trivalent chromium passivation system to passivate zinc alloy coatings. While the article presents an interesting study, there are a few areas that require attention:

·       The introduction and abstract could more effectively highlight the novelty of the research.

·       In Figure 1, the authors should add arb. units as the label for the y-axis (intensity).

·       To aid readers who are not familiar with the topic, the conclusion section should be expanded.

·       It may be beneficial for the authors to include Raman measurements to provide a more detailed characterization of the samples.

Moderate editing of English language.

Reviewer 2 Report

In this article, the author has investigated the silicate passivation system and a trivalent chromium passivation system for the passivation of zinc alloy coating. The passivation film was formed by coordination between negatively charged SiO2- or SiO2 micelles and Zn2+ having a lower corrosion current and higher corrosion potential than the trivalent chromium passivation film. The author has done appreciable work, but the manuscript needs to go through minor revision for enhancement of its quality. Kindly find my comments for the minor revision of the article:

1.     Provide more context on the objective and significance of the study in the abstract.

2.     Clarify the methodology used for passivating the zinc alloy coating.

3.     Elaborate on the potential applications or implications of the findings.

4.     Consider mentioning any limitations or further research that could be explored based on the results.

5.     Expand the introduction to provide a more comprehensive overview of the research area and the need for passivation technologies. Provide clearer transitions between the different studies and their findings.

6.     Include a statement on the novelty or contribution of the proposed silicate passivation formula.

7.     Provide more details on the materials and methods used, including the rationale for selecting specific parameters. Provide specific details on the equipment and techniques used for microstructure characterization.

8.     Consider revising the language for clarity and conciseness.

9.     Clarify the purpose and significance of the surface morphology and composition analysis.

10.  Provide a more concise explanation of the XRD results, emphasizing the lack of shift and new peaks in the diffraction patterns.

11.  Explain the presence of surface defects, dendritic structures, and large particles in the unpassivated sample.

12.  Highlight the improved surface morphology and absence of dendritic structures after trivalent chromium passivation.

13.  Discuss the uniform and flat surface morphology, tight tissue structure, and uniform coverage of the silicate passivated sample.

14.  Summarize the corrosion resistance results and emphasize the lower corrosion current density and higher corrosion resistance of the passivated samples compared to the unpassivated sample.

15.  Clarify the importance of the EIS curves in evaluating the corrosion resistance of the film.

16.  Emphasize the increase in low-frequency capacitive reactance arc, improved high-frequency area, and increased corrosion impedance after passivation.

17.  Explain the presence of small cracks in the passivation film and their impact on the equivalent circuit diagram used for fitting.

18.  Provide a clear comparison of the polarization resistance (Rp) values for each sample, emphasizing the higher Rp value for the silicate passivated sample, indicating better corrosion resistance and protection ability.

19.  Include the specific duration of the salt spray test (72 hours) and clarify that it was conducted in a humid and hot environment.

20.  Clearly explain the formation of white rust due to the electrochemical reaction between chloride ions and the zinc coating.

21.  Emphasize that the presence of white rust indicates a compromised passivation film and reduced corrosion resistance.

22.  Highlight the findings that the silicate passivation sample had the least white rust, indicating the best corrosion resistance, followed by the trivalent chromium passivation sample.

23.  Connect the results of the salt spray test with the SEM observations, emphasizing the reduction in surface defects and specific surface area due to passivation.

24.  Conclude the section by summarizing the major findings related to the passivation film's effect on crystal structure, the formation of silicate passivation film, and the improved corrosion resistance of the silicate passivation system compared to traditional trivalent chromium passivation.

25.  Kindly expand the literature by citing recent articles such as: Current Nanoscience 18, no. 2 (2022): 203-216 http://dx.doi.org/10.2174/1573413717666210216120741; Applied Sciences 13, no. 2 (2023): 730 https://doi.org/10.3390/app13020730

Minor editing of English language required

Round 2

Reviewer 1 Report

I recommend the publication of this work. 

 Minor editing of English language required.